# Unveiling Hidden Risks: Intentional Molecular Screening for Sexually Transmitted Infections and Vaginosis Pathogens in Patients Who Have Been Exclusively Tested for Human Papillomavirus Genotyping

**DOI:** 10.3390/microorganisms11112661

**Published:** 2023-10-30

**Authors:** Fabiola Hernández-Rosas, Manuel Rey-Barrera, Flavio Hernández-Barajas, Claudia Rangel-Soto, Mariana Socorro García-González, Shumeyker Susmith Franco-González, Mercedes Piedad de León-Bautista

**Affiliations:** 1Biomedical Engineering Faculty, Engineering Division, Anahuac Queretaro University, Querétaro 76246, Mexico; fabiola.hernandezro@anahuac.mx; 2Research Center, Anahuac Queretaro University, Querétaro 76246, Mexico; 3Cambrico Biotech, 41015 Sevilla, Spain; 4Translational Medicine, Vanguard and Technology Transfer Sector, Human Health Department, Central ADN Laboratories, Morelia 58280, Mexico; 5Escuela de Medicina, Universidad Vasco de Quiroga, Morelia 58090, Mexico

**Keywords:** STI, multiplex PCR detection, coinfections, vaginosis pathogens, human papillomavirus

## Abstract

Human papillomavirus (HPV) is the most prevalent sexually transmitted infection (STI) worldwide, with popular screening methods including the Papanicolaou test and HPV genotyping. However, in clinical practice, coinfections with other pathogens are often underestimated. Therefore, our study aims to describe the prevalence of STIs and vaginosis in urogenital samples from patients who had been tested exclusively for HPV genotyping. Methods: This analytical, prospective, cross-sectional study included 408 males and females. Eligible participants had positive and negative HPV genotyping test results and agreed to early detection or had HPV antecedents. They provided the same urogenital samples used for HPV detection and, through our multiplex in-house PCR assay, we screened for *Candida* spp., *Ureaplasma* spp., *Trichomonas vaginalis*, *Neisseria gonorrhoeae*, *Chlamydia trachomatis*, herpes simplex virus 1 and 2 (HSV), *Mycoplasma* spp., molluscum contagiosum virus (MCV), *Treponema pallidum*, *Haemophilus* spp., *Staphylococcus aureus*, and *Klebsiella* spp. The subsequent statistical analysis aimed to reveal correlations between HPV genotypes and the identified pathogens. Results: Of the participants, 72.1% (n = 294) tested positive for HPV genotypes. HR-HPV (high-risk HPV) genotypes comprised 51 (8.1%), 66 (7.1%), and 58 (6.1%). *Haemophilus* spp., *Ureaplasma* spp., *Candida* spp., *Staphylococcus aureus*, and *Mycoplasma spp*. frequently co-occurred with HPV infection (*p* < 0.05). Gender-based variations were notorious for *Ureaplasma* spp., *Mycoplasma* spp., and MCV (*p* < 0.05). Coinfections were prevalent (43.9%), with a positive HPV result elevating the risk for *Trichomonas vaginalis*, *Mycoplasma* spp., *Staphylococcus aureus*, HSV, and MCV (OR > 1, *p* < 0.05). HPV 16 correlated with HSV and *Ureaplasma* spp., while HPV 6 was linked with HSV and MCV (*p* < 0.05). Conclusions: This screening strategy uncovered significant coinfections and associations between HPV genotypes and pathogens, underscoring the importance of routine screening to explore clinical implications in urogenital health.

## 1. Introduction

Human papillomavirus (HPV) is the most common sexually transmitted infection (STI) in the world and, due to its relationship with oncogenic processes, PAP tests or HPV genotype identification are present in routine clinical practice. Much evidence supports the theory that infection at urogenital sites increases the risk of acquiring other pathogens and developing diseases, either by predisposing the microenvironment or complicating or increasing HPV-induced carcinogenesis, causing concerning clinical complications [1,2]. However, cotesting for HPV along with other STIs and bacterial vaginosis is underused, limiting the possibility of precise detection.

The detection of STIs and other pathogenic microorganisms, such as HPV and the bacterium responsible for bacterial vaginosis, is crucial to public health and patient wellbeing. However, in clinical practice, simultaneous testing for these pathogens is often underestimated. This lack of attention given to simultaneous detection can hinder the precise identification of coinfections, which, in turn, can adversely affect the diagnosis, treatment, and prevention of these infections.

Molecular detection methods based on molecular biology and polymerase chain reaction (PCR) have emerged as an essential tool for screening for multiple infections. These advanced techniques enable the identification of specific genetic material from different pathogens in a single sample, simplifying the detection process and significantly increasing the accuracy. Furthermore, molecular methods offer additional advantages, such as a greater sensitivity and specificity, the ability to identify specific viral or bacterial strains, and the early detection of infections that may be asymptomatic in their initial stages [3]. In particular, the multiplex PCR technique enables the simultaneous detection of STI pathogens in urogenital samples via a single reaction, saving time and resources [4]. Furthermore, it is cost-effective, enhances the sensitivity and specificity, facilitates the identification of coinfections, and assists doctors in making informed decisions, leading to more effective treatment and, thus, improving patient care [3].

Today, in medical practice, the identification of *Chlamydia trachomatis*, *Neisseria gonorrhoreae*, *Mycoplasmas*, *Ureaplasmas*, and HPV is routinely performed; nevertheless, in some cases, the presence of other viruses, like molluscum contagiosum virus (MCV), which has been detected in patients with positive and negative HPV test results [5], can give rise to a clinical misdiagnosis or complicate clinical manifestations. Therefore, as physicians, to expand the diagnostic window, we should use differential diagnostic tests alongside HPV to test for conditions such as herpes simplex virus 1 and 2 (HSV), *Klebsiella* spp., and *Staphylococcus aureus* infections, among others.

Hence, our first study goal was to encourage an intentional STI and bacterial vaginosis screening campaign in patients who had been exclusively tested for HPV genotyping, with positive and negative results, through the analysis of their cervicovaginal, urethral, external genitalia, penis, and genital wart samples (the same as had been used for the HPV assay) to detect *Candida* spp., *Ureaplasma* spp., *Trichomonas vaginalis*, *Neisseria gonorrhoeae*, *Chlamydia trachomatis*, HSV, *Mycoplasma* spp., MCV, *Treponema pallidum*, *Haemophilus* spp., *Staphylococcus aureus*, and *Klebsiella* spp., using a multiplex PCR in-house validated method. Furthermore, we evaluated the risk of coinfections in HPV-positive patients and assessed their potential impact on the sexual health and fertility of the study population.

## 2. Materials and Methods

### 2.1. Study Population and Selection Criteria

This analytical, prospective, and cross-sectional study included 408 females and males and was conducted from December 2021 to June 2022. The participants were aged from 20 to 80 years old. The requirements for participation were the donation of the same samples used for HPV genotyping and agreement with early detection or HPV antecedents (Table 1) to simultaneously screen for different microbes: *Candida* spp., *Ureaplasma* spp., *Trichomonas vaginalis*, *Neisseria gonorrhoeae*, *Chlamydia trachomatis*, HSV, *Mycoplasma* spp., MCV, *Treponema pallidum*, *Haemophilus* spp., *Staphylococcus aureus*, and *Klebsiella* spp.

### 2.2. Ethical Statement

The study was conducted in compliance with the ethical principles supported by the Declaration of Helsinki and the current health laws of Mexico. The study protocol was approved by the medical and ethical research committee at Central ADN (project number SH-202021HPVST). All the participants signed informed consent as part of the protocol established by the ethical research committee. The information and results were kept confidential. The collected data were coded with a serial number, and the identification of all participants was kept anonymous. The samples were collected and processed by trained personnel in the Molecular Diagnostic Laboratory Central ADN in Morelia, Michoacán, Mexico.

### 2.3. Specimen Collection

In the present study, we used the same extracted DNA for the HPV genotyping test from urogenital samples of the patients. A trained physician completed the clinical examination and collected samples from different anatomical sites: cervicovaginal, urethral, external genitalia, penis, and genital warts, as was previously described [6].

Briefly, to collect the urethral and external genitalia samples, two sterile ultrafine flexible swabs HydraFlock^®^ (Puritan, Guilford, CT, USA; 25-3306-H) were used. Cervical and vaginal samples were obtained by scraping the epithelium with a nylon cytobrush (Puritan, Guilford, CT, USA; 2199). Genital warts were excised as part of treatment and dissected using sterilized scalpel blades (KLS martin, Burgenland, Austria; 10-150-10-04). Every specimen was stored in Universal Transport Medium (Copan Diagnostics, Brescia, Italy; 350C) at room temperature and processed in the molecular diagnostic laboratory Central ADN in Morelia, Michoacán, Mexico. All staff engaged in collecting, transporting, and processing the specimens received proper training in handling protocols. Also, each sample was labeled with an alphanumeric code to ensure identification and appropriate monitoring of the technical process in our lab.

### 2.4. DNA Extraction

The DNA for this study was extracted using Instagene Matrix (Bio-Rad, Hercules, CA, USA; 7326030), with minor changes to the manufacturer’s instructions. Briefly, 200 μL of the sample was transferred to 200 μL tubes. Then, the samples were centrifuged at 14,000 rpm for 1 min. The pellet was washed twice with PBS-Tween 20. Subsequently, 50 µL of Instagene Matrix was added to the pellet and incubated at 56 °C for 30 min and 100 °C for 8 min in a Veriti™ 96-Well Fast Thermal Cycler (Thermo Scientific, Waltham, MA, USA; 4375786). For genital warts, we used the Exgene™ Cell SV protocol (GeneAll, Seoul, Republic of Korea; 106-101). Finally, the supernatants were carefully transferred to a new sterile Eppendorf tube and were stored at −20 °C until their use. DNA quality, purity, and quantity were evaluated using a NanoDrop™ 2000/2000c Spectrophotometer (Thermo Scientific, Waltham, MA, USA; ND-2000). The 260/280 ratio of absorbance for pure DNA was ~2.0.

### 2.5. HPV Genotypes Detection

The HPV genotypes detection was performed using the fHPV typing™ kit (Genomed Biotech, London, UK; mlg.hpv.100) to simultaneously detect 6, 11, 16, 18, 31, 33, 35, 39, 45, 51, 52, 56, 58, 59, 66, and 68 genotypes, by multiplex fluorescent PCR targeting the E6 and E7 regions of the HPV genome.

A final 10 μL reaction mix was prepared by adding 5 μL of the master mix reaction, 1.5 μL of BSA 10%, 1 μL of the primers, and 2.5 μL of extracted DNA. The PCR program consisted of a HotStart activation step at 95 °C for 5 min, followed by 35 cycles of denaturation at 95 °C for 60 s, annealing at 66 °C for 30 s, and extension at 72 °C for 30 s, and final extension step at 72 °C for 10 min. For the following 34 cycles, the annealing temperature decreased by 0.3 °C each. The PCR reactions were performed using a Veriti™ 96-Well Fast Thermal Cycler (Thermo Scientific, Waltham, MA, USA; 4375786). Then, the fragments were resolved in an Automated Genetic Analyzer ABI 3500 (Applied Biosystems, Waltham, MA, USA; 4426479) using a GeneScan 500 LIZ Size Standard (Waltham, MA, USA; 4322682) and HD-Formamide.

Electrophoresis reactions were used in a final volume of 5.5 μL with 4.5 μL of reaction mix + 1 μL of PCR product. The electropherogram generated by an ABI 3500 Genetic Analyzer was analyzed using the software GeneMapper ID-X v1.3 (Applied Biosystems, Waltham, MA, USA; 4473491) as described by the supplier.

### 2.6. Primer Design

The DNA sequences of MCV, *Treponema pallidum*, *Haemophilus* spp., *Staphylococcus aureus*, and *Klebsiella* spp., were obtained from GenBank (https://www.ncbi.nlm.nih.gov/genbank/ accessed on 21 September 2021) and the accession numbers are listed in Table 1. This panel was named the Complementary STI Panel.

Every primer pair was designed with nondimerizing primer set combinations using the MPprimer software (http://www.premierbiosoft.com/primerplex/index.html accessed on 4 January 2022) [7]. The primer sequences for PCR assay of bacterial and viral pathogens are described in Appendix A.

For the detection of *Candida* spp., *Ureaplasma* spp., *Trichomonas vaginalis*, *Neisseria gonorrhoeae*, *Chlamydia trachomatis*, HSV 1 and 2, and *Mycoplasma* spp., we used the methodology previously reported [8]. This panel was named the Basic STI Panel.

**Table 1 microorganisms-11-02661-t001:** GenBank accession numbers to design the multiplex PCR assay for the Complementary STI Panel.

Pathogen	Primer Name	Gene	Amplicon Size (pb)	Gene Bank Access	Reference
Molluscum Contagiosum Virus	M1_MCV_F1 M1_MCV_R1	Major envelope protein	269	KX214741.1	In-house
HSV	M1_HSV_F1 M1_HSV_R1	DNA pol	205	MH697529.1 MH697449.1	In-house
*Treponema pallidum*	Primer KO Primer KO4	DNA pol	260	CP073552.1	[9]
*Haemophilus* spp.	M1_Hadu_F1 M1_Hadu_R1	16s rRNA	154	LT694117.1	In-house
*Staphylococcus aureus*	M1_Stau_F1 M1_Stau_R1	RNA pol	123	CP076105.1	In-house
*Klebsiella* spp.	M1_Klgr_F1 M1_Klgr_R1	PhoE	100	CP073373.1	In-house

Abbreviations: HSV, herpes simplex virus; 16S rRNA, 16S ribosomal RNA; kDa: kilodaltons; RNA Pol, RNA polymerase D; PhoE, phosphoporin PhoE.

### 2.7. Positive Control Strains, Sequence Analysis, and Validation

As positive controls, well-characterized clinical samples of MCV, *Treponema pallidum*, *Haemophilus* spp., *Staphylococcus aureus*, and *Klebsiella* spp., were kindly shared by Hospital de la Mujer, Michoacán, Mexico.

To ensure the specificity and to find a possible cross-identification of the primers, an end-point PCR was performed, and the amplified products were sequenced using forward and reverse primers at Macrogen Inc. (Seoul, Republic of Korea). The sequences were then blasted against available sequences in the GenBank to confirm the identity of the microorganisms.

### 2.8. Positive Control Constructs

Once we confirmed the specificity of the primers designed, the DNA fragments were cloned into the pGEM vector to generate the constructs for the Complementary STI Panel; pGEM/HA, pGEM/MCV, pGEM/T. pallidum, pGEM/S. aureus and pGEM/*Klebsiella* spp., as positive synthetic controls. For the Basic STI Panel, we used the constructs mentioned in our previous study [8].

The plasmids were added to *Escherichia coli DH-5α* strain and purified with alkaline lysis. The inserts were verified by a PCR reaction and confirmed by sequencing analysis at Macrogen Inc. (Seoul, Republic of Korea). The sequences were blasted against available sequences in the GenBank to identify the microorganisms.

### 2.9. PCR Assays Conditions

The single PCR reactions were carried out using QIAGEN^®^ Multiplex PCR Kit (Qiagen, Hilden, Germany; 206145), in a final volume of 10 μL prepared with PCR master mix buffer [1X], and 2 μL (100 ng/μL) of total nucleic acid extraction. Once the single target PCR system succeeded, we performed the in-house multiplex PCR assay, mixing the primers for the Complementary STI Panel (Appendix A). Due to the variability of primers Tm and to increase the specificity, we performed a touch-down PCR, as described previously [8].

To demonstrate the presence of genomic human material, human actin primers were used 5′ACCGAGCGCGGCTACAG3′ and 5′ CTTAATGTCACGCACGATTTCC3′ forward and reverse (ShineGene Molecular Biotech, Shanghai, China), respectively. The positive samples were analyzed by sequencing at Macrogen Inc. (Seoul, Republic of Korea). The sequences were blasted against available sequences in the GenBank to identify and corroborate the microorganism.

Finally, the combination of Basic and Complementary STI Panels was suitable to identify *Candida* spp., *Ureaplasma* spp., *Trichomonas vaginalis*, *Neisseria gonorrhoeae*, *Chlamydia trachomatis*, HSV, *Mycoplasma* spp., MCV, *Treponema pallidum*, *Haemophilus* spp., *Staphylococcus aureus*, and *Klebsiella* spp. in the same sample used to detect HPV genotypes.

### 2.10. Statistical Analysis

IBM SPSS version 28 was used for our descriptive and inferential statistics (IBM Statistics, Armonk, NY, USA; 6413381). Quantitative data with a normal distribution were described as mean SD (standard deviation), while qualitative data were expressed as frequencies and percentages. The chi^2^ test examined the relationship between HPV genotypes and infections with microorganisms that cause STIs and/or vaginosis. *p* < 0.05 was chosen as the statistical significance threshold. Additionally, a conditional logistic regression was carried out to evaluate the chance of coinfection in the presence of a positive HPV test and to provide odds ratios (ORs) with 95% confidence intervals.

## 3. Results

Data from 408 participants were analyzed. Initially, we analyzed the anthropometric and clinical characteristics of the patients. We found that the average age of the patients was 30.2 ± 10.7 years; 53.9% were female and 46.1% were male. The main reasons for the clinical analysis were early detection (90.9%), condylomas (3.2%), and HPV history (2.7%) (Table 2).

We investigated the frequency of high- and low-risk HPV genotypes in both sexes. The overall prevalence of high-risk (HR) HPV was 33.3%, and low-risk (LR) HPV was 37.5% in both men and women (Table 2). We found that the most frequent HR-HPV genotypes in women were 51 (8.1%), 66 (7.1%), 58 (6.1%), 52 (3.9%), and 16 (3.4%). In men, the most frequent HR-HPV genotypes were 51 (5.6%), 66 (3.4%), 59 (3.2%), and 16 (3.2%). The most frequent genotype in both genders was 6 (18.1% and 11.8%, respectively) (Table 3).

We examined the frequency distribution of HPV genotypes in both genders, and we found a significant association between the sex of the patient and the frequency of low-risk HPV genotypes 6 and 11 (*p* < 0.01) (Table 3).

For the numerous detections of the microorganisms that cause STI and vaginosis, two internal STI panels (Basic and Complementary Panels) based on multiplex PCR were used. The panels were validated previously in the laboratory, evaluating the robustness and reproducibility of the results. Figure 1a,b show the results of the Basic and the Complementary STI Panel with the participant’s samples. Figure 1c demonstrates well-characterized samples and the pool of plasmids (as a synthetic positive control) in the Complementary STI Panel. As stated in the methodology, the expected band for each pathogen was sequenced to verify the accuracy of the findings. The Basic STI Panel was standardized, according to a prior study [8].

Furthermore, we investigated the prevalence of coinfection (HPV with other microorganisms) with the most common etiological agents of STI in both men and women. We identified *Haemophilus* spp. (32.4%), *Ureaplasma* spp. (24.8%), *Candida* spp. (11.8%), *Staphylococcus aureus* (7.1%), and *Mycoplasma* spp. (6.9%) as the most frequently associated pathogens with HPV in our patients. Among women, the most common pathogens were *Ureaplasma* spp. (15.9%), *Haemophilus* spp. (15.4%), and *Mycoplasma* spp. (5.6%), while among men, they were *Haemophilus* spp. (16.9%), *Ureaplasma* spp. (8.8%), and *Candida* spp. (6.6%) (Table 4; Figure 2b).

In addition, we examined the association between the frequency of the analyzed pathogens and the sex of the patient. We found significant differences between men and women for the following pathogens: *Ureaplasma* spp. (*p* = 0.015), *Mycoplasma* spp. (*p* = 0.002), and MCV (*p* = 0.023) (Table 4). Specifically, *Ureaplasma* spp. and *Mycoplasma* spp. were more prevalent in women, while MCV was more common in men (Table 4; Figure 2b).

The median age of patients with a positive HPV result was 34 years, and for those with a negative HPV, the result was 33 years. The median age of cases that tested positive for STI-causing bacteria was 34 years old, compared to 32 years for cases that tested negative. Figure 3 displays the distribution of age ranges by sex.

Out of the total number of patients analyzed, 72.1% had some HPV genotype, of which 27.0% had multiple HPV genotype infections. A total of 43.9% of the patients were positive for HPV and some other STI-causing pathogen. Moreover, we found that there was no significant risk for a patient with a positive HPV result to present an infection by some other associated pathogen (*p* > 0.05) (Table 5).

From several anatomical areas, biological samples for STI and HPV genotyping analysis were collected. Table 6 shows the frequency of each anatomical site in men and women, as well as the anatomical sites positive for HPV or STI. In general, the cervix (50.5%) and urethra (44.7%) were the most frequently sampled sites in both men and women, respectively. Additionally, 12.0% and 30.4% of the results were positive in these anatomical regions.

We explore the possibility that HPV infection may be a risk factor for infection by some of the pathogens studied. Thus, we found statistical differences when comparing HPV cases with the detection of *Mycoplasma* spp. (*p* = 0.035), *Staphylococcus aureus* (*p* = 0.009), and MCV (*p* = 0.022) (Table 7).

Moreover, HPV infection was found to be a risk factor for *Trichomonas vaginalis* (OR = 1.389; 95% CI = 1.307–1.476), *Mycoplasma* spp. (OR = 3.439; 95% CI = 1.170–11.622), *Staphylococcus aureus* (OR = 5.663; 95% CI = 1.324–24.218), HSV (OR = 1.393; 95% CI = 1.310–1.481), and MCV (OR = 1.406; 95% CI = 1.320–1.497) (Table 7).

Additionally, we examined the likelihood that having a specific HPV genotype (high or low probability) may increase the likelihood of receiving a positive STI test result. As shown in Table 8, none of the HPV genotypes that were most frequently observed in our study are substantially linked (*p* > 0.05) with the chance of receiving a positive STI test.

Finally, we investigated the relationship between several HPV genotypes and their potential role as STI risk factors. We found that genotype 16 was significantly associated with HSV infections (OR = 15.160; 95% CI = 2.049–112.165; *p* = 0.000) and *Ureaplasma* spp. (OR = 0.228; 95% CI = 0.053–0.980; *p* = 0.031) (Table 9). Genotype 6 was significantly associated with HSV and MCV (*p* = 0.048 and *p* = 0.000). Genotype 6 was a risk factor for MCV (OR = 8.423; 95% CI = 2.276–31.173). Genotype 66 was a risk factor for HSV 1 and 2 infections (OR = 8.854; 95% CI = 1.215–64.536; *p* = 0.010) (Table 9).

## 4. Discussion

The HPV infection is commonly acquired through sexual contact, and, in most cases, the immune system resolves the infection. However, eventually, the virus remains chronic, a concerning fact due to its association with oncogenic processes and the potential development of lesions or other health issues. Several approaches have been used to deal with the confusing HPV nature, including routine cervical cancer screening and HPV vaccination for advised populations. The current cervical cancer screening guidelines recommend the Papanicolaou test and HPV genotyping [10], and they demonstrate significant benefits such as decreased morbidity and mortality [11]. Unfortunately, the development of any oncogenic process is multifactorial, and STI is one of the strongest facts to acquire an HPV infection. Therefore, we invited patients whose doctors sent urogenital samples for the sole purpose of identifying and genotyping HPV to take part in our intentional screening campaign to reveal other pathologies or infections, supporting the expansion of the current screening strategies and drastically reducing STI contagious among men and women, as was proposed by the Sustainable Development Goals (SDGs) [12]. Due to the main objective of this study being to detect microorganisms in patients beyond the scope of HPV genotyping testing, we developed an in-house multiplex PCR assay to determine pathogens related to ulcerative lesions: MCV, *Treponema pallidum*, HSV, *Staphylococcus aureus*, *Klebsiella* spp., and *Haemophilus* spp. In addition, in the second line, we sought the most prevalent pathogens, such as *Candida* spp., *Ureaplasma* spp., *Trichomonas vaginalis*, *Neisseria gonorrhoeae*, *Chlamydia trachomatis*, and *Mycoplasma* spp.

In our study population, we observed that the prevalence of HR-HPV and LR-HPV were similar, 33% and 37.5%, respectively, and the most common HPV genotypes were, in decreasing order, 6, 11, 51, 66, 58, 59, 16, 52, and 39. Although different methods were used to determine HPV genotypes, the results were equivalent to those of earlier studies. The more frequent HPV genotypes included 66, 52, and 51, in addition to 16 [13,14,15,16]. One of the most intriguing details was that our findings were comparable to those of a Mexican study [17]. In conclusion, the support for developing vaccines with targets routinely discovered worldwide is the most representative influence of all these similarities. 

Not all HPV panels typically include LR-HPV genotypes; however, interest in these genotypes has grown due to their potential relevance to carcinogenic processes not only in the cervix but also associated with vaginal intraepithelial neoplasia (VaIN) [13,18,19]. Here, we found that 6 LR-HPV was the most frequently observed in both genders, with a higher presence in men. The HIM study reported the frequency of this genotype and variations primarily in normal genital skin [20]. These facts can be explained by two factors: first, nononcogenic HPV has been shown to have a lower viral load in men, and second, the immune system reacts less strongly to prevent HPV infection [21].

Because the participants of the study were sent exclusively to detect any HPV genotype and there were no other apparent clinical signs or symptoms to suspect having any other infection, the results were surprising. Furthermore, in addition to the high rates of *Ureaplasmas* spp. and *Mycoplasmas* spp. in females, the other common pathogen was MCV, mainly in men. In the case of *Ureaplasmas* and *Mycoplasmas*, it is well studied that these microorganisms are acquired from the first instance of sexual intercourse, and during many years, it was believed they were innocuous; nevertheless, those mollicutes are being correlated with poor pregnancy, perinatal outcome, and infertility among men and women [22,23].

MCV was the most frequent virus detected in male participants, and there is a lack of evidence for HPV-negative lesions in men where the pathogen present is different, as reported previously [24]. The coexistence of several viruses is commonly seen in patients with high-risk sexual behavior or with a low immune response [25,26].

One of the main goals of our study was to identify the pathogens present in HPV-positive samples. We found that pathogens such as *Chlamydia trachomatis*, *Neisseria gonorrhoeae*, HSV, and *Trichomonas vaginalis* were detected. These findings are one of the reasons why we conducted this study, as these pathogens often present subclinically, and patients may continue to develop complications and sexual health outcomes such as infertility and a higher risk of acquiring other infections, such as HPV and HIV. Our results demonstrate the importance of routinely screening patients for these pathogens.

In our population study, although our data did not show a significant difference, it was noteworthy that HPV-positive samples had a high frequency of positive STI results. Moreover, HPV multiple infections were more frequent in patients with positive results for other STIs. These cellular events are frequently seen by the imbalance of the microenvironment brought on by the dynamic molecular interaction of coexisting bacteria in a particular mucosa, a fact that is commonly thought to be the origin of these cellular processes [27,28]. For instance, *Trichomonas vaginalis* was not a critical pathogen detected in our population compared to other people in Latin America [29,30]. In addition, scientific evidence links the microbiota with other molecular immune markers, such as interleukins, neutrophils, macrophages, and others [31].

This project demonstrated that we should prioritize the development of screening tests and apply them intentionally to improve clinical outcomes, prevalence/incidence, and other relevant sanitary information to provide concrete answers for patients. Innovation is necessary to develop new technologies and processes and improve upon and tailor existing ones to different target audiences, environments, or objectives. The most important conclusion is that having solid data on STIs will help targeted programs be more accurate and effective in adopting or adapting services to benefit a more significant number of people, strengthening strategic information systems, and ensuring that data are gathered and used ethically to help communities requiring the cooperation of civil society. Additionally, it will be crucial to establish affordable STI diagnostic tests and more capable laboratories in Mexico and Latin America [32].

Understanding the STI epidemic requires knowing when, how, and among whom new infections are happening. It also entails determining the conditions that facilitate the transmission of STIs or hinder their control and the use of treatments that can help. This information enables prevention programs, treatment, and care to be prioritized and given special attention. Finally, our research findings can influence clinical practice and public health regulations to enhance patient outcomes and stop the spread of STIs. 

## Figures and Tables

**Figure 1 microorganisms-11-02661-f001:**
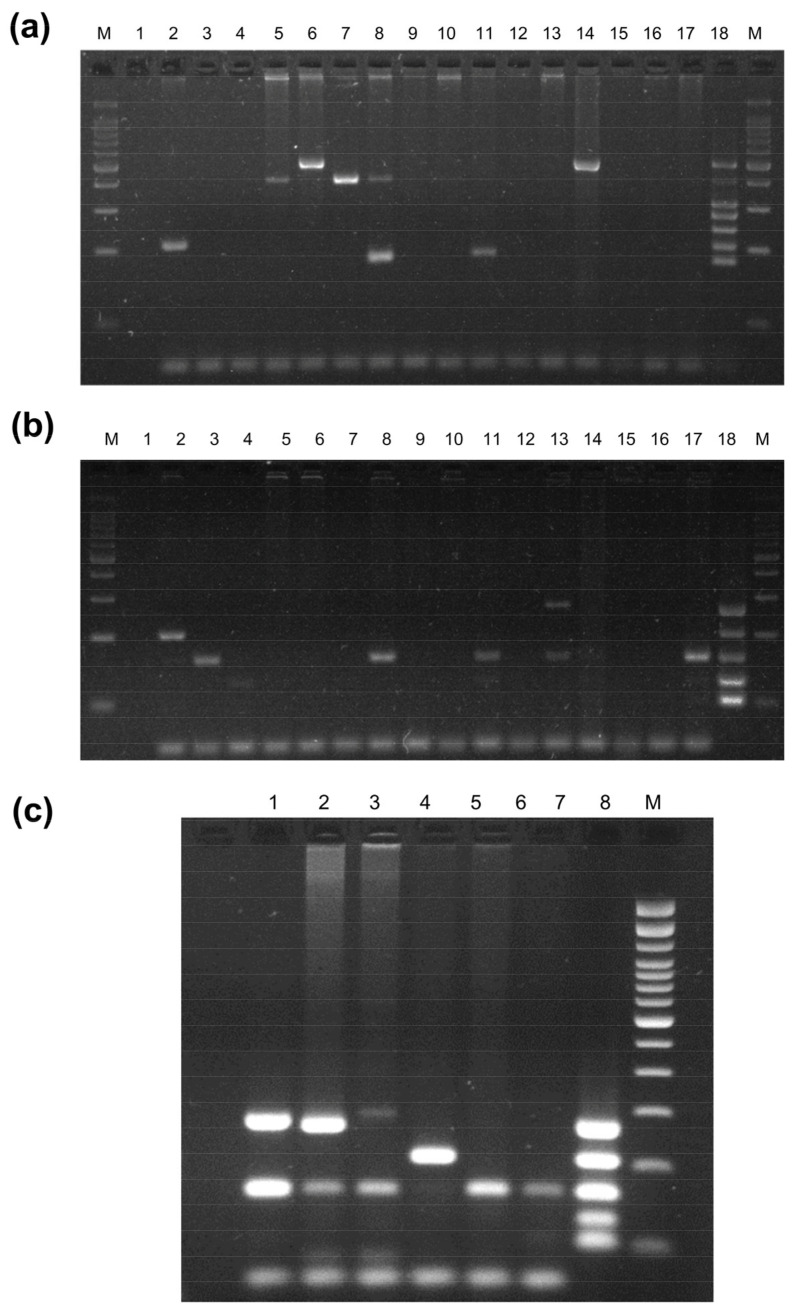
Multiplex PCR assay results and in-house validation. (**a**) Basic STI Panel and (**b**) Complementary STI Panel results. Gel electrophoresis (2.5% agarose) of PCR products from the participants to detect *Candida* spp. (473 bp)*; Ureaplasma* spp. (397 bp); *Trichomonas vaginalis* (308 bp); *Neisseria gonorrhoeae* (278 bp); *Chlamydia trachomatis* (241 bp); HSV (205 bp); *Mycoplasma* spp. (184 bp); MCV (269 bp); *Treponema pallidum* (262 bp); *Haemophilus* spp. (157 bp); *Staphylococcus aureus* (123 bp); and *Klebsiella* spp. (100 bp). M, Ladder 100 bp marker (Invitrogen, Waltham, MA, USA); (1) negative control (molecular grade water). Samples: (2) HSV and *Haemophilus* spp., (3) *Haemophilus* spp., (4) *Staphylococcus aureus*; (5) *Ureaplasma* spp., (6) *Candida* spp., (7) *Ureaplasma* spp., (8) *Ureaplasma* spp., *Mycoplasma* spp. *and Haemophilus* spp., (9, 10, 12, 15, 16) negative; (11) *Mycoplasma* spp., *Haemophilus* spp. and *Staphylococcus aureus*; (13) MCV and *Haemophilus* spp., (14) *Candida* spp., (17) *Haemophilus* spp., *Staphylococcus aureus and Klebsiella* spp. (18) Pool of cloning plasmids as a positive control. (**c**) Complementary STI Panel of well-characterized samples and plasmids pool as a synthetic control. Gel electrophoresis (2.5% agarose) of PCR products from well-characterized samples to detect MCV (269 bp); *Treponema pallidum* (262 bp); *Haemophilus* spp. (157 bp); *Staphylococcus aureus* (123 bp); *and Klebsiella* spp. (100 bp). (8) Pool of cloning plasmids as a positive control. M, Ladder 100 bp marker (Invitrogen, Waltham, MA, USA); (1) negative control (molecular grade water). Samples: (2) MCV and *Haemophilus* spp., (3) *Treponema pallidum* and *Haemophilus* spp., (4) MCV and *Haemophilus* spp., (5) HSV; (6, 7) *Haemophilus* spp., *Haemophilus* spp., and *Klebsiella* spp. The band of ~65 bp corresponds to human actin beta as an internal control.

**Figure 2 microorganisms-11-02661-f002:**
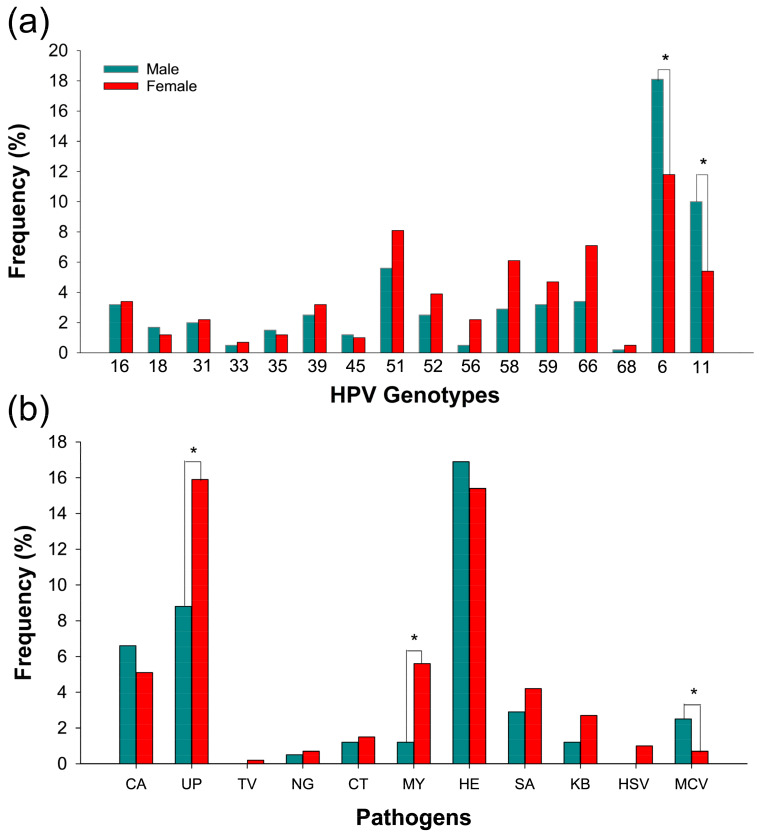
Frequency of microorganisms causing STI in men and women. (**a**) Frequency of HPV genotypes. (**b**) Frequency of pathogens detected in coinfection with HPV. Abbreviations: CA, *Candida* spp.; UP, *Ureaplasma* spp.; TV, *Trichomonas vaginalis*; NG, *Neisseria gonorrhoeae*; CT, *Chlamydia trachomatis*; MY, *Mycoplasma* spp.; HE, *Haemophilus* spp.; SA, *Staphylococcus aureus*; KB, *Klebsiella* spp.; *HSV*, herpes simplex virus 1 and 2; MCV, molluscum contagiosum virus. The ANOVA test was used to compare between groups. A statistically significant value was indicated as * *p* < 0.05.

**Figure 3 microorganisms-11-02661-f003:**
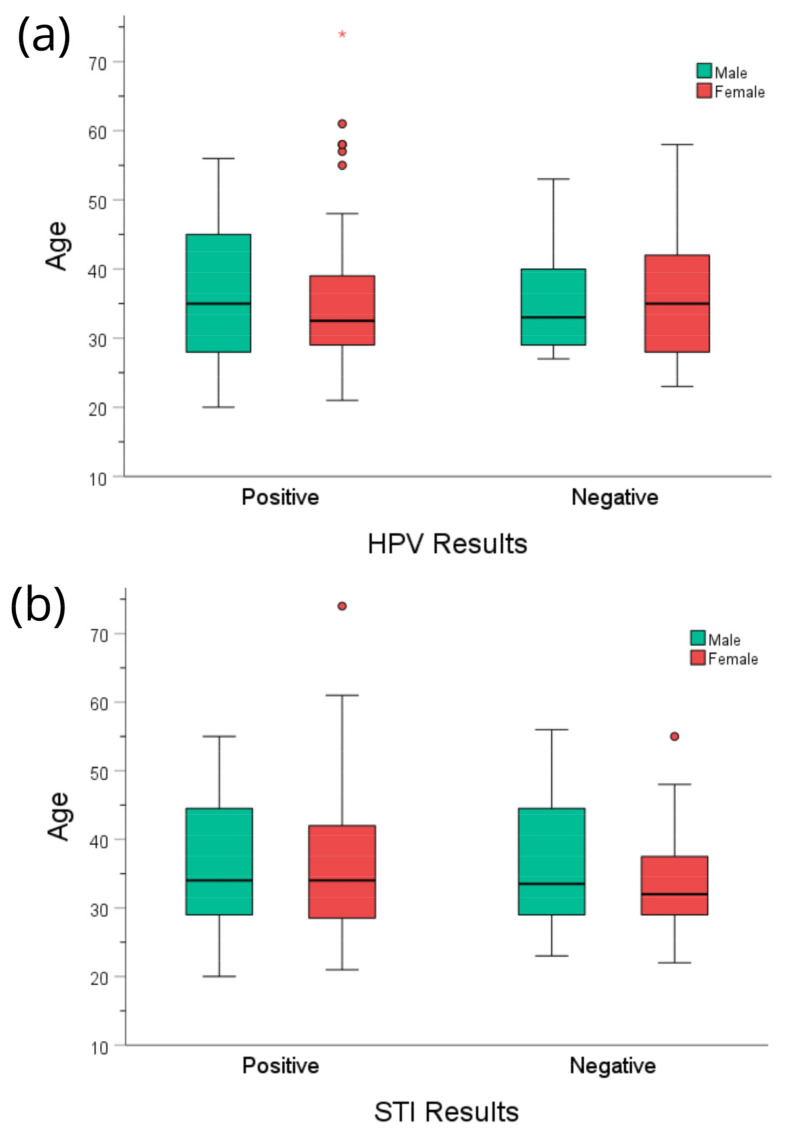
Age distribution of patients of both sexes with positive results for (**a**) the genotypes of HPV and (**b**) pathogens causing STI or vaginosis.

**Table 2 microorganisms-11-02661-t002:** Anthropometric and clinical characteristics of the study population (n = 408).

Variables	Data
Age (years) ^a^	30.2 ± 10.7
Gender of patients	
Female	220 (53.9)
Male	188 (46.1)
Reason for clinical analysis ^b^	
HPV history	11 (2.7)
Condylomas	13 (3.2)
Early detection	371 (90.9)
NIC 1	2 (0.5)
Genital ulcers	2 (0.5)
Genital warts	9 (2.2)

^a^ Data are shown as means and standard deviation. ^b^ Data are shown as frequencies and percentages.

**Table 3 microorganisms-11-02661-t003:** Frequency of HPV genotypes according to sex (n = 408).

HPV Genotype	Total		Male		Female		*p*-Value
	n	%	n	%	n	%	
HR-HPV	136	33.3	62	15.2	74	18.1	0.888
16	27	6.6	13	3.2	14	3.4	0.823
18	12	2.9	7	1.7	5	1.2	0.387
31	17	4.2	8	2.0	9	2.2	0.934
33	5	1.2	2	0.5	3	0.7	0.784
35	11	2.7	6	1.5	5	1.2	0.568
39	23	5.6	10	2.5	13	3.2	0.797
45	9	2.2	5	1.2	4	1.0	0.564
51	56	13.7	23	5.6	33	8.1	0.418
52	26	6.4	10	2.5	16	3.9	0.421
56	11	2.7	2	0.5	9	2.2	0.060
58	37	9.1	12	2.9	25	6.1	0.081
59	32	7.8	13	3.2	19	4.7	0.519
66	43	10.5	14	3.4	29	7.1	0.060
68	3	0.7	1	0.2	2	0.5	0.657
LR HPV	153	37.5	90	22.1	63	15.4	0.000
6	122	29.9	74	18.1	48	11.8	**0.000 ***
11	63	15.4	41	10.0	22	5.4	**0.001 ***

Abbreviations: HPV, human papillomavirus; HR-HPV, high-risk HPV; LR-HPV, low-risk HPV. Significant difference between men and women was established by Xi^2^ test. * A *p*-value < 0.05 was considered significant.

**Table 4 microorganisms-11-02661-t004:** Evaluation of coinfections in female and male patients with HPV positive test (n = 179).

Pathogens	Total		Male		Female		*p*-Value
	n	%	n	%	n	%	
*Candida* spp.	48	11.8	27	6.6	21	5.1	0.132
*Ureaplasma* spp.	101	24.8	36	8.8	65	15.9	**0.015 ***
*Trichomonas vaginalis*	1	0.02	0	0.0	1	0.2	0.355
*Neisseria gonorrhoeae*	1	0.2	2	0.5	3	0.7	0.657
*Chlamydia trachomatis*	11	2.7	5	1.2	6	1.5	0.966
*Mycoplasma* spp.	28	6.9	5	1.2	23	5.6	**0.002 ***
*Treponema pallidum*	0	0.0	0	0.0	0	0.0	-
*Haemophilus* spp.	132	32.4	69	16.9	63	15.4	0.083
*Staphylococcus aureus*	29	7.1	12	2.9	17	4.2	0.598
*Klebsiella* spp.	16	3.9	5	1.2	11	2.7	0.225
HSV 1/2	4	1.0	0	0.0	4	1.0	0.063
MCV	13	3.2	10	2.5	3	0.7	**0.023 ***

Abbreviations: MCV, molluscum contagiosum virus; HSV, herpes simplex virus 1 and 2. Significant difference between men and women was established by Xi^2^ test. * A *p*-value < 0.05 was considered significant.

**Table 5 microorganisms-11-02661-t005:** HPV and STI results (n = 408).

HPV Results	Total		STI Positive		STI Negative		OR	95% CI	*p*-Value
	n	%	n	%	n	%			
HPV Positive Test									
Yes	294	72.1	179	43.9	115	28.2	1.053	0.677–1.637	0.819
No	114	27.9	68	16.7	46	11.3	-	-	
HPV multiple infection									
Yes	110	27.0	65	15.9	45	11.0	0.921	0.590–1.438	0.716
No	182	44.6	116	28.4	298	73.0	-	-	

Abbreviations: HPV, human papillomavirus; STI, STD; OR, odds ratio; CI, confidence interval. Significant difference between men and women was established by Xi^2^ test. A *p*-value < 0.05 was considered significant.

**Table 6 microorganisms-11-02661-t006:** Frequency of positive results of STI according to anatomical site of specimen collection (n = 408).

Anatomical Site	Total		Male		Female		HPV+/STI+	
	n	%	n	%	n	%	n	%
External genitalia *	19	4.7	9	4.8	10	4.5	14	3.4
wart	26	6.4	24	12.8	2	0.9	14	3.4
Urethra	86	21.1	84	44.7	2	0.9	49	12.0
BG	7	1.7	7	1.7	-	-	2	0.5
Foreskin	3	0.7	3	0.7	-	-	3	0.7
Penis	54	13.2	54	13.2	-	-	36	8.8
Anus	4	1.0	4	1.0	0	0.0	2	0.5
Vagina	3	0.7	-	-	3	0.7	3	0.7
Cervix	206	50.5	-	-	206	50.5	124	30.4

Abbreviations: STI, STD; balanoprepucial groove. * Includes vulva, scrotum, and pubis.

**Table 7 microorganisms-11-02661-t007:** Frequency of pathogens associated with HPV results and risk factors (n = 408).

Pathogens	Total		HPV Positive		HPV Negative		OR	95% CI	*p*-Value *
	n	%	n	%	n	%			
*Candida* spp.	48	11.8	40	9.8	8	2.0	2.087	0.945–4.608	0.064
*Ureaplasma* spp.	101	24.8	66	16.2	35	8.6	0.653	0.403–1.059	0.083
*T. vaginalis*	1	0.2	1	0.2	0	0.0	**1.389**	**1.307–1.476**	0.553
*N. gonorrhoeae*	3	0.7	2	0.5	1	0.2	0.774	0.069–8.620	0.835
*C. trachomatis*	11	2.7	6	1.5	5	1.2	0.454	0.136–1.519	0.189
*Mycoplasma* spp.	28	6.9	25	6.1	3	0.7	**3.439**	**1.170–11.62**	**0.035**
*Haemophilus* spp.	132	32.4	91	22.3	41	10.0	0.798	0.506–1.259	0.331
*S. aureus*	29	7.1	27	6.6	2	0.5	**5.663**	**1.324–24.21**	**0.009**
*Klebsiella* spp.	16	3.9	12	2.9	4	1.0	1.170	0.369–3.706	0.789
HSV 1/2	4	1.0	4	1.0	0	0.0	**1.393**	**1.310–1.481**	0.211
MCV	13	3.2	13	3.2	0	0.0	**1.406**	**1.320–1.497**	**0.022**

Abbreviations: HPV, human papillomavirus; MCV, molluscum contagiosum virus; HSV 1/2, herpes simplex virus 1 and 2; OR, odds ratio; CI, confidence interval. * Xi^2^ test was used to compare between groups. A *p*-value < 0.05 was considered significant.

**Table 8 microorganisms-11-02661-t008:** STI results associated with HPV genotypes (n = 408).

HPV Genotype	Total		STI Positive		STI Negative		OR	CI 95%	*p*-Value
	n	%	n	%	n	%			
HR-HPV	136	33.3	80	19.6	56	13.7	0.898	0.590–1.367	0.616
16	27	6.6	13	3.2	14	3.4	0.583	0.267–1.276	0.173
39	23	5.6	15	3.7	8	2.0	1.237	0.512–2.987	0.800
51	56	13.7	36	8.8	20	4.9	1.203	0.669–2.163	0.537
52	26	6.4	14	3.4	12	2.9	0.746	0.336–1.657	0.471
58	37	9.1	23	5.6	14	3.4	1.078	0.537–2.163	0.832
59	32	7.8	21	5.1	11	2.7	1.267	0.594–2.704	0.540
66	43	10.5	29	7.1	14	3.4	1.397	0.714–2.733	0.328
LR HPV	153	37.5	98	24.0	55	13.5	1.268	0.838–1.917	0.261
6	122	29.9	74	18.1	48	11.8	1.007	0.653–1.554	0.975
11	63	15.4	41	10.0	22	5.4	1.258	0.718–2.203	0.423

Abbreviations: HPV, human papillomavirus; HR-HPV, high-risk HPV; LR-HPV, low-risk HPV; STI, STD; OR, odds ratio; CI, confidence interval. Significant difference between men and women was established by Xi^2^ test. A *p*-value < 0.05 was considered significant.

**Table 9 microorganisms-11-02661-t009:** STI results associated with HPV results (n = 408).

HPV Results	STI Results	STI Results	OR	CI 95%	*p*-Value
HPV 16					
Genotype 16	HSV Positive	HSV Negative	OR	CI 95%	*p*-Value
Positive	2 (0.5)	25 (6.1)	15.160	2.049–112.165	0.000
Negative	2 (0.5)	379 (92.9)	-	-	
HPV 16					
Genotype 16	*Ureaplasma* spp. Positive	*Ureaplasma* spp. Negative	OR	CI 95%	*p*-Value
Positive	2 (0.5)	25 (6.1)	0.228	0.053–0.980	0.031
Negative	99 (24.3)	282 (69.1)	-	-	
HPV 6					
Genotype 6	MCV Positive	MCV Negative	OR	CI 95%	*p*-Value
Positive	10 (2.5)	112 (27.5)	8.423	2.276–31.173	0.000
Negative	3 (0.7)	283 (69.4)	-	-	
HPV 6					
Genotype 6	HSV Positive	HSV Negative	OR	CI 95%	*p*-Value
Positive	3 (0.7)	119 (29.2)	7.185	0.740–69.772	0.048
Negative	1 (0.2)	285 (69.9)	-	-	
HPV 66					
Genotype 66	HSV Positive	HSV Negative	OR	CI 95%	*p*-Value
Positive	2 (0.5)	41 (10.0)	8.854	1.215–64.536	0.010
Negative	2 (0.5)	363 (89.0)	-	-	

Abbreviations: HPV, human papillomavirus; STI, STD; OR, odds ratio; CI, confidence interval. Xi^2^ test was used to compare between groups. A *p*-value < 0.05 was considered significant.

## Data Availability

The data presented in this study are available on request from the corresponding author. The data are not publicly available due to the study protocol (project number SH-202021HPVST), which stipulates that the information and results are confidential.

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
