# Peer review of "Unveiling Hidden Risks: Intentional Molecular Screening for Sexually Transmitted Infections and Vaginosis Pathogens in Patients Who Have Been Exclusively Tested for Human Papillomavirus Genotyping"

_microorganisms, 2023, doi:10.3390/microorganisms11112661_

Round 1

Reviewer 1 Report

This manuscript aimed to the prevalence of STI and bacterial vaginosis alongside Human Papillomavirus in urogenital samples.

This manuscript addresses important point in the field.

title.

Define STI in the title

Abstract

In general, I think the abstract need to be rewritten to improve the clarity and the flow.

Specific comment:

Using a validated multiplex PCR method, significant correlations between HPV genotypes and pathogens were analyzed: Is this method or finding. How you analyzed the significant correlation.

Line 22: can you write the rest of spp instead of etc.

Line 23:  Our results showed HPV-positive were found in 72.1% (n=294 patients). Need to be rewritten.

Line 24: define HR-HPV genotypes.

Line 26: were frequently associated, with what?

Line 29: define HSV, and MCV

Introduction:

The introduction section needs more information to cover the intentional STI and bacterial vaginosis screening. How these are performed and why it is important, what is the difference between them and the other method of diagnosis.

The objectives of the study and why it is important should be mentioned in the introduction section.

Methods

Line 62 Should you include the patient’s metadata?

Line 82 What kind of samples were used for DNA extraction.

Results and discussion are well written and well presented.

I would suggest combine figure 1 and figure 2 together.

Author Response

September 29th, 2023

Rebuttal Letter to microorganisms-2625411 (First round)

Unveiling Hidden Risks: Intentional Molecular Screening of Sexually Transmitted Infections and Vaginosis Pathogens in Patients who were Exclusively Tested for HPV Genotyping

Fabiola Hernández-Rosas, Manuel Rey-Barrera, Flavio Hernández-Barajas, Claudia Rangel Soto, Mariana Socorro García-González, Shumeyker Susmith Franco González, Mercedes Piedad de León-Bautista

Dear Editor, and dear reviewers,

We would like to thank you for your comments and suggestions, which allowed us to improve our work. In the revised manuscript (MS Microorganisms-2625411-Corrections 29-09-23), we hope to solve all the issues raised. In this document, we answer all the questions asked by the reviewers. Comments are shown in bold font, followed by our answer/comment in normal font. The major corrections/changes in the manuscript are displayed in red font.

REVIEWER 1:

Define STI in the title

Response: Thank you for pointing this out. We made some slight changes to the title. Instead of "Unveiling Hidden Risks: Intentional Molecular Screening of STI and Vaginosis Pathogens in Patients who Were Exclusively Tested for HPV Genotyping Test", we have “Unveiling Hidden Risks: Intentional Molecular Screening of Sexually Transmitted Infections and Vaginosis Pathogens in Patients who were Exclusively Tested for HPV Genotyping”

Abstract:

In general, I think the abstract needs to be rewritten to improve the clarity and flow.

Response: We agree with this comment. Therefore, we have rewritten the abstract. Lines 30-51.

Specific comment:

Using a validated multiplex PCR method, significant correlations between HPV genotypes and pathogens were analyzed: Is this method or finding? How do you analyze the significant correlation?

Response: We thank you for these observations that let us clarify the analysis. The Chi2 test was used to correlate positive HPV results with positive results for individual pathogens. In addition, a conditional logistic regression test was performed to obtain the ORs at the 95% confidence interval (CI) and establish the risk that patients with a positive HPV test are positive for some other pathogen. These analyses were performed in SPSS version 28 software as was indicated in the methodology.

Line 22: can you write the rest of spp instead of etc.

Response: Agree. To solve this relevant observation, the error was corrected in the abstract.

Line 23: Our results showed HPV-positive were found in 72.1% (n=294 patients). Need to be rewritten.

Response: Thank you for this pertinent observation. The error was corrected in the text with “Of the participants, 72.1% (n=294) were positive for HPV genotypes”. Line 43.

Line 24: define HR-HPV genotypes.

Response: We thank you for the detail. We inserted the definition of High Risk-HPV. Line 44.

Line 26: were frequently associated, with what?

Response: We appreciate this observation that let us clarify the concept. The error was corrected in the text “Haemophilus spp., Ureaplasma spp., Candida spp., Staphylococcus aureus, and Mycoplasma spp. frequently co-occurred with HPV infection (p<0.05)”. Lines 43-45.

Line 29: define HSV, and MCV

Response: The comment is very punctual. We inserted the definitions of Herpes Simplex Virus 1 and 2 (HSV) and Molluscum Contagiosum Virus (MCV). Lines 39 and 40. 

Introduction:

The introduction section needs more information to cover the intentional STI and bacterial vaginosis screening. How these are performed and why are they important, what is the difference between them and the other method of diagnosis?

Response: We agree with this comment. Therefore, we thank you for pointing this out. The information was inserted in the text. Line 64-79 

The objectives of the study and why it is important should be mentioned in the introduction section.

Response: We agree with this comment. We paraphrased the goals of our study. Lines 87-95.

Methods

Line 62 Should you include the patient’s metadata?

Response: Thank you for this pertinent observation. Concerning this fact, we have inserted a Table (Table 2). We highlight that we did not collect more clinical information because our campaign focused on intentional detection, primarily for early detection or HPV antecedents. Lines 101-103 and 202-227.

Line 82 What kind of samples were used for DNA extraction?

Response: It is a very good note. We added “ urogenital samples” in the sentence “...we used the same extracted DNA for the HPV genotyping test from urogenital samples of the patients…” Lines 124 and 125.

The results and discussion are well-written and well-presented.

Response: Thanks for these relevant remarks. Therefore, we revised both sections and added a very specific point associated with Vaginal intraepithelial neoplasia (VaIN) (DOI: 10.1097/LGT.0000000000000732). Lines 385 and 386.

I would suggest combining Figure 1 and Figure 2.

Response: We embrace the relevant information. We combined the figures. Lines 250-266.

We thank the editor and the reviewers for all your valuable comments and suggestions, and we are confident that with all these improvements we hope to fulfill all the requirements to publish our manuscript in your journal. 

Sincerely,

Mercedes Piedad de León Bautista

Central ADN Laboratories

Universidad Vasco de Quiroga

dramercedespiedad@gmail.com 

September 29th, 2023

Reviewer 2 Report

I would add an additional table about baseline population information

It is not clear how the population was selected: were the people recruited sequentially or were they selected? The high prevalence of STIs in your study population may mean that people accessed primarily for an STIs and then tested for HPV. This is crucial and must be clarified, it is a crucial potential bias

I would suggest citing DOI: 10.1097/LGT.0000000000000732 for updated evidence on management of VaIN.

Thank you for your precious work

Moderate editing

Author Response

September 29th, 2023

Rebuttal Letter to microorganisms-2625411 (First round)

Unveiling Hidden Risks: Intentional Molecular Screening of Sexually Transmitted Infections and Vaginosis Pathogens in Patients who were Exclusively Tested for HPV Genotyping

Fabiola Hernández-Rosas, Manuel Rey-Barrera, Flavio Hernández-Barajas, Claudia Rangel Soto, Mariana Socorro García-González, Shumeyker Susmith Franco González, Mercedes Piedad de León-Bautista

Dear Editor, and dear reviewers,

We would like to thank you for your comments and suggestions, which allowed us to improve our work. In the revised manuscript (MS Microorganisms-2625411-Corrections 29-09-23), we hope to solve all the issues raised. In this document, we answer all the questions asked by the reviewers. Comments are shown in bold font, followed by our answer/comment in normal font. The major corrections/changes in the manuscript are displayed in red font.

REVIEWER 2:

I would add a table about baseline population information

Response: Thank you for this pertinent observation. Concerning this fact, we have inserted a Table with the anthropometric and clinical characteristics of the study population (Table 2). We highlight that we did not collect more clinical information because our campaign focused on intentional detection, primarily for early detection. Lines 231-232.

It is not clear how the population was selected: were the people recruited sequentially or were they selected? The high prevalence of STIs in your study population may mean that people accessed primarily for STIs and then tested for HPV. This is crucial and must be clarified, it is a crucial potential bias.

Response: We thank the Reviewer for this observation that clarifies the quality of our work. Concerning this fact, we paraphrased the study population and selection criteria. Lines 104-110.

I would suggest citing DOI: 10.1097/LGT.0000000000000732 for updated evidence on the management of VaIN.

Response: Thanks for these relevant observations. We added a very specific point associated with the Vaginal intraepithelial neoplasia (VaIN) (Reference No. 20). Lines 385 and 386.

Thank you for your precious work

Response: Thank you for your time and attention to this matter and, for encouraging our efforts.

Moderate editing

Response: We thank the Reviewer for this observation that improves the quality of our work. We have checked and revised the entire document to ensure English Quality. 

We thank the editor and the reviewers for all your valuable comments and suggestions, and we are confident that with all these improvements we hope to fulfill all the requirements to publish our manuscript in your journal. 

Sincerely,

Mercedes Piedad de León Bautista

Central ADN Laboratories

Universidad Vasco de Quiroga

dramercedespiedad@gmail.com 

September 29th, 2023

Round 2

Reviewer 2 Report

I am fine with the revised manuscript